# Intestinal Farnesoid X Receptor Modulates Duodenal Surface Area but Does Not Control Glucose Absorption in Mice

**DOI:** 10.3390/ijms24044132

**Published:** 2023-02-18

**Authors:** Jiufang Yang, Theo H. van Dijk, Martijn Koehorst, Rick Havinga, Jan Freark de Boer, Folkert Kuipers, Tim van Zutphen

**Affiliations:** 1Department of Pediatrics, University of Groningen, University Medical Center Groningen, 9700RB Groningen, The Netherlands; 2Department of Laboratory Medicine, University of Groningen, University Medical Center Groningen, 9700RB Groningen, The Netherlands; 3European Research Institute for the Biology of Ageing (ERIBA), University of Groningen, University Medical Center Groningen, 9700RB Groningen, The Netherlands; 4Faculty Campus Fryslân, University of Groningen, 8911CE Leeuwarden, The Netherlands

**Keywords:** glucose absorption, intestinal FXR, duodenum, GS3972

## Abstract

Bile acids facilitate the intestinal absorption of dietary lipids and act as signalling molecules in the maintenance of metabolic homeostasis. Farnesoid X receptor (FXR) is a bile acid-responsive nuclear receptor involved in bile acid metabolism, as well as lipid and glucose homeostasis. Several studies have suggested a role of FXR in the control of genes regulating intestinal glucose handling. We applied a novel dual-label glucose kinetic approach in intestine-specific FXR^−/−^ mice (iFXR-KO) to directly assess the role of intestinal FXR in glucose absorption. Although iFXR-KO mice showed decreased duodenal expression of hexokinase 1 (*Hk1*) under obesogenic conditions, the assessment of glucose fluxes in these mice did not show a role for intestinal FXR in glucose absorption. FXR activation with the specific agonist GS3972 induced *Hk1*, yet the glucose absorption rate remained unaffected. FXR activation increased the duodenal villus length in mice treated with GS3972, while stem cell proliferation remained unaffected. Accordingly, iFXR-KO mice on either chow, short or long-term HFD feeding displayed a shorter villus length in the duodenum compared to wild-type mice. These findings indicate that delayed glucose absorption reported in whole-body FXR^−/−^ mice is not due to the absence of intestinal FXR. Yet, intestinal FXR does have a role in the small intestinal surface area.

## 1. Introduction

Apart from their classical functions in fat absorption, bile acids have been recognised as hormone-like molecules that regulate glucose, lipid and energy metabolism [1]. Early evidence showing a link between bile acids and glucose metabolism comes from studies using the bile acid sequestrants to show improved glycaemic control in patients with type 2 diabetes [2,3,4]. It has been hypothesised that the beneficial effect of bile acid sequestrants on glucose metabolism involves Farnesoid X receptor (FXR) signalling [5]. This nuclear receptor is expressed in many tissues and organs, including the liver, intestine, kidneys, adrenal glands, white adipose tissue and immune cells [6]. FXR is a major regulator of bile acid metabolism but also strongly modulates various physiological processes by up- or downregulating target genes in response to bile acids and specific agonists [1]. The activation of FXR mediates glucose and lipid metabolism, especially in the organs taking part in the enterohepatic cycle of bile acids, i.e., liver and intestine. Several synthetic FXR agonists have been developed and are in clinical trials, such as obeticholic acid (OCA) for the treatment of primary sclerosing cholangitis and Cilofexor for non-alcoholic steatohepatitis (NASH) [7,8,9,10]. A role for bile acids and FXR has been proposed for the most potent intervention for type 2 diabetes, i.e., bariatric surgery [11,12,13]. Activation of FXR specifically in the intestine with the gut-restricted agonist fexaramine induces circulating fibroblast growth factor 15/19 (FGF15/19) levels, thereby altering hepatic bile acid synthesis and bile acid composition, leading to increased energy expenditure and improved metabolic profiles [14,15]. Apart from the liver, FXR signalling also affects peripheral insulin sensitivity in adipose tissue and skeletal muscle, indicating the important role of FXR in whole-body glucose homeostasis through various direct and indirect mechanisms that may involve other bile-responsive receptors [16,17].

FXR is expressed throughout the small intestine, i.e., in the duodenum, jejunum and ileum [18]. While FXR in the ileum regulates the reabsorption of bile acids, its control over the transintestinal cholesterol excretion pathway may occur more upstream [19]. Synthetic agonists, especially the non-steroidal agonists, also likely activate intestinal FXR more proximally compared to bile acids that are mostly taken up in the ileum. Contrasting findings were presented on the mechanisms of pharmaceutical modulation of intestinal FXR. Beneficial effects of the gut-restricted FXR agonist fexaramine were shown to rely on lithocholic acid (LCA)-producing bacterial species in the gut that lead to increased activation of TGR5 by the potent agonist LCA to stimulate intestinal glucagon-like peptide-1 (GLP-1) secretion, inducing adipose tissue browning and, ultimately, improving insulin sensitivity [20]. In contrast, other studies have shown that the activation of intestinal FXR has a detrimental effect on glucose homeostasis and energy expenditure in response to obesity [21,22]. For instance, the use of the FXR antagonist glycine-β-muricholic acid, that inhibits FXR signalling exclusively in the intestine, was proposed to improve the metabolic parameters in mouse models of obesity through an intestinal FXR–ceramide axis [22]. The intestinal FXR/GLP-1 pathway has also been shown as another FXR-mediated pathway in glucose metabolism: the inhibition of FXR activity in L-cells by colesevelam induces the expression of proglucagon, subsequently promoting GLP-1 production and, hence, improving glucose metabolism [23]. Taken together, these results indicate that FXR signalling is complex, and the activity of FXR in the intestine can exert effects on whole-body metabolism through various mechanisms. Besides controlling metabolic pathways, the activation of FXR has been shown to improve the intestinal structure and integrity in several disease models [18,24,25]. The regulation of intestinal architecture may therefore represent another possibility of metabolic control through modulating the intestinal surface area.

Intestinal glucose absorption is the initial step of prandial glucose metabolism. An increase in the expression of the major apical transporter, sodium-dependent glucose transporter 1 (SGLT1), leads to increased glucose absorption in obese mice [26]. Proximal intestinal glucose absorption is also accelerated in obese humans [27]. In contrast, humans with gastrointestinal exclusion by Roux-en-Y gastric bypass show decreased intestinal glucose absorption and improved whole-body glucose metabolism [28]. Our previous study has shown that FXR affects glucose kinetics by modulating intestinal glucose absorption in the duodenum, the main site of glucose absorption. In whole-body FXR^−/−^ mice, the appearance of glucose in the blood appeared to be delayed and to accumulate as glucose-6-phosphate in enterocytes, requiring dephosphorylation to allow release into the blood [29]. However, this mechanism has not been studied further, and in view of the pleiotropic actions of FXR, the use of the total FXR^−/−^ model might not allow the assessment of a direct role for FXR in the intestine.

Here, we aimed to study the role of intestinal FXR in glucose handling directly by analysing glucose absorption in an intestine-specific FXR^−/−^ (iFXR-KO) mouse model using a novel dual-label glucose kinetic approach. We hypothesised that the control of FXR on the expression of glucose transporters and/or kinases, as well as the intestinal architecture, might contribute to the intestinal glucose absorption capacity.

## 2. Results

### 2.1. FXR Deficiency in Intestine Does Not Lead to Delayed Glucose Absorption

To determine whether FXR directly modulates glucose handling in the intestine, mice lacking FXR specifically in enterocytes (iFXR-KO mice) were employed. IFXR-KO mice had similar body weights and small intestine lengths compared to wild-type mice (Appendix A). Consistent with previous findings [30], ileal expression of *Fgf15* was significantly lower in iFXR-KO mice compared to controls, indicating impaired FXR-FGF15 signalling (Appendix A). In line, hepatic expression of cytochrome P450 family 7 subfamily A member1 (*Cyp7a1*) was significantly higher in iFXR-KO mice, whereas cytochrome P450 family 8 subfamily B member1 (*Cyp8b1*), reported to be predominantly controlled by hepatic FXR, was unaffected [30] (Appendix A).

Whereas whole-body FXR^−/−^ mice were shown to have higher hexokinase 1 (*Hk1*) gene expression in the first segment of the small intestine, only a minor trend to higher expression was observed for this gene in the first of the five segments of the small intestine (i.e., the duodenum) of iFXR-KO mice compared to wild-type mice (Figure 1a). There was no significant difference in intestinal hexokinase 2 (*Hk2*) expression either between wild-type and iFXR-KO mice (Appendix A). The expression of the major glucose transporters sodium-dependent glucose cotransporter 1 (*Sglt1*) and the intracellular glucose-6-phosphate transporter (*G6pt*) all tended to be higher in the duodenum of iFXR-KO mice compared to wild-type and glucose transporter 2 (*Glut2*) was significantly increased in the first segment (Figure 1b, Appendix A). To quantify the glucose absorption, a novel dual-label glucose kinetic approach was developed (Appendix A). Tracer kinetics, however, showed identical glucose appearance rates and recovery of the oral tracer between wild-type and iFXR-KO mice (Figure 1c,d), indicating that, despite the changes in gene expression of glucose transporters, the temporal dynamics of intestinal glucose absorption were similar between the two groups.

Next, we analysed the morphology of the small intestine in wild-type and iFXR-KO mice on chow diet to determine whether there were any changes in absorptive surface area. IFXR-KO mice had a significantly shorter villus length in the duodenum compared to that of wild-type mice (Figure 1e,f, 549.59 μm vs. 488.68 μm, 12%). Yet, no significant difference in duodenal crypt length was observed between the two groups (Figure 1g). Additionally, the villus length was solely increased in the duodenum, whereas no change in length was observed in the jejunum or ileum compared to controls (Figure 1h,i). Together, these gene expression data, morphology and kinetic analysis indicate that despite effects on intestinal surface and gene expression of glucose-related genes, deficiency of FXR specifically in the intestine does not affect the glucose absorption rate. 

### 2.2. Long-Term HFD Exposure Does Not Affect Glucose Absorption in iFXR-KO Mice

As exposure to a HFD affects glucose metabolism in mice [31], we next tested intestinal glucose absorption in wild-type and iFXR-KO mice after 10-week HFD feeding to determine whether glucose absorption in mice lacking intestinal FXR may be affected under these conditions. Duodenal expression of *Hk1* was significantly lower in iFXR-KO mice after 10-week HFD while expression of glucose transporters was not affected (Figure 2a). Similar to chow-fed conditions, however, glucose absorption kinetics remained highly similar between iFXR-KO mice and controls. The appearance kinetics of orally administered glucose were virtually identical and also the recovery of the oral tracer did not differ between both groups (Figure 2b,c).

Citrulline levels were significantly increased after 6 weeks of HFD feeding in wild-type mice indicating increased enterocyte mass. However, no significant changes were found between genotypes when mice were fed either chow, 6-week HFD or 10-week HFD (Figure 2d). In line with the observations on chow diet, intestinal morphology under obesogenic conditions showed significantly lower villus length in the duodenum of iFXR-KO mice compared to wild-type (Figure 2e,f). For other parts of the intestine however, villus length between the two groups was similar (Figure 2f). Additionally, there was no significant difference in crypt length in the duodenum and jejunum of wild-type and iFXR-KO mice. Yet, a small but significant increase in crypt length was observed in the ileum of iFXR-KO mice compared to controls under obesogenic conditions (Figure 2g).

### 2.3. Long-Term HFD Exposure Perturbs Whole-Body Glucose Metabolism to the Same Extend in Wild-Type and iFXR-KO Mice

The kinetic analyses of glucose fluxes during the glucose absorption test also allow identification of changes in whole-body glucose homeostasis. During the test, blood glucose concentrations in wild-type and iFXR-KO mice on chow diet showed virtually identical dynamics with a peak level at around 15 min (16.30 mM in wild-type mice and 16.45 mM in iFXR-KO mice) (Figure 3a). A separate cohort of mice was fed a HFD for 10 weeks. Body weight gain was very similar in both wild-type and iFXR-KO mice (Figure 3b). Notably, the peak values of blood glucose concentration in wild-type and iFXR-KO mice after HFD were higher than their corresponding values under chow conditions (19.78 mM in wild-type mice and 17.58 mM in iFXR-KO mice) and blood glucose concentrations showed delayed dynamics. However, we did not observe differences between genotypes (Figure 3c). Glucose clearance rates were identical between the two genotypes under both chow and HFD conditions (Figure 3d). However, the obese HFD-fed groups displayed a decreased clearance rate compared to lean chow-fed mice. Additionally, no significant differences were found in the degree to which the hepatic glucose production was suppressed by the glucose bolus between wild-type and iFXR-KO mice on both diets, although this suppression was clearly impaired in HFD-fed mice compared to chow-fed animals (Figure 3e). In line, insulin sensitivity also remained highly comparable between iFXR-KO mice and controls, whereas long-term HFD exposure similarly reduced insulin sensitivity equally in both wild-type and iFXR-KO mice (Figure 3f). HFD feeding is associated with ectopic lipid accumulation in mice. Histology of liver showed that there were indeed more lipid droplets in livers from mice fed with HFD. Yet again, both genotypes appeared to be similarly affected (Figure 3g). This result was different from previous studies showing that intestinal FXR promotes hepatic lipid accumulation [22,32]. To investigate whether bile acid homeostasis was affected in iFXR-KO mice, bile acid composition in plasma and bile was analysed. Similar to previous findings [30], there was no significant difference in plasma bile acid concentrations between wild-type and iFXR-KO mice. However, iFXR-KO mice tended to have slightly higher percentages of taurocholic acid, and these mice had significantly less muricholates in plasma compared to wild-type mice after 10-week HFD feeding. No differences in bile acid composition were observed in bile after 10-week HFD feeding either (Figure 3h). Together, these results confirm that long-term HFD leads to impaired glucose metabolism, reduced insulin sensitivity as well as accumulation of hepatic triglycerides in mice. However, the deficiency of FXR in the intestine does not impact on these changes within the applied timeframe. 

### 2.4. Activation of FXR by GS3972 Does Not Alter Glucose Absorption in Wild-Type Mice

To determine whether pharmacological activation of intestinal FXR has an impact on intestinal glucose absorption, FXR activation with the agonist GS3972 was applied for further analysis of the role of FXR in glucose absorption. As the onset exposure to excess dietary fat already affect multiple parameters that advance metabolic disease development later on, including glucose tolerance [33], we focused on the early stages of obesity development [34]. The expression of both *Hk1* and *Hk2* was already significantly lower in the first segment of the small intestine of iFXR-KO mice compared to wild-type under short-term HFD conditions (Figure 4a,b). The effect of GS3972 treatment on glucose absorption was analysed in both wild-type and iFXR-KO mice during the onset of obesity-induced metabolic derangements after a one-week HFD. GS3972 induced expression of Fgf15 and ileal bile acid-binding protein (*I-babp*) in the ileum of wild-type mice but not in iFXR-KO mice (Appendix A). In livers of both wild-type and iFXR-KO mice treated with GS3972, expression of the FXR target gene small heterodimer partner (*Shp*) was significantly increased while the expression of *Cyp7a1* and *Cyp8b1* was significantly decreased (Appendix A–e), indicating that the agonist activates FXR in both intestine and liver.

As described above, expression of *Hk1* in the duodenum of iFXR-KO mice was significantly lower than wild-type mice after one-week HFD feeding, while activation of FXR increased the expression of *Hk1* in duodenum of wild-type mice, but not in iFXR-KO mice (Figure 4c). A similar trend was detected for duodenal expression of *Hk2*; however, this did not reach significance (Figure 4d). There was no difference in expression of *Sglt1* and other glucose transporters among treated and untreated groups (Figure 4e and Appendix A–h). Glucose absorption kinetics showed that, apart from a trend in the initial phase, activation of FXR by GS3972 did not significantly impact intestinal glucose absorption rates, as evident from similar appearance rate and cumulative recovery of orally-administrated glucose (Figure 4f,g). There were also no significant differences in whole-body glucose clearance rates or suppression of endogenous glucose production between the groups, indicating that FXR activation did not prevent or aggravate initial development of glucose intolerance and insulin resistance (Figure 4h,i).

### 2.5. Activation of FXR by GS3972 Increases Villus Length in Duodenum

Despite that several studies in disease models identified a role for FXR in recovery of intestinal morphology and structure [24,35,36], enterocyte mass did not appear to be altered by activation of FXR during short-term HFD feeding, as indicated by comparable plasma citrulline levels in treated and untreated wild-type as well as iFXR-KO mice (Figure 5a). The morphology of the duodenum, however, was slightly but significantly affected by GS3972-induced FXR activation as quantitative analysis showed that duodenal villus length was increased upon GS3972 treatment in wild-type (Figure 5b, 523.64 μm vs. 573.56 μm, 9% increase). In iFXR-KO mice however, GS3972 treatment did not affect villus length (Figure 5b and Appendix A). Consistent with our earlier results (Figure 1), villus length was significantly shorter in the duodenum of untreated iFXR-KO mice compared to controls. Crypt length was not influenced by activation of FXR (Figure 5c). In line with the comparable total enterocyte mass between treatment groups and genotypes, this difference of villus length was not observed in other parts of the small intestine (jejunum and ileum) in either wild-type mice treated with GS3972 compared to controls or in iFXR-KO mice, except for a small but significant increase in jejunal crypt length in iFXR-KO mice compared to untreated controls (Appendix A).

We finally explored whether this increased villus length in duodenum by activation of FXR was due to an increased proliferation rate of intestinal epithelial cells. The new epithelial cells differentiate from stem cells at the base of intestinal crypts and gradually migrate upward to the villus tips. To track proliferative cells and their descendants along the crypt–villus axis, Brdu was injected 24 h before termination. Brdu-positive cells were detected in the duodenum of all mice from every group. However, we did not observe any difference in terms of proliferation rate between treated and untreated groups or between iFXR-KO and wild-type mice (Figure 5d,e), indicating that the increased villus length in the duodenum is not due to increased cell proliferation within crypts. As pro-inflammatory cytokines may also have an impact on enterocytes by regulating proliferation and differentiation of epithelial cells [37] and FXR is able to modulate this response [25,38], we also measured the pro-inflammatory cytokines in wild-type and iFXR-KO mice treated with and without GS3972. However, no differences in expression of inflammatory marker genes in the duodenum of both treated as well as untreated wild-type and iFXR-KO mice were detected (Appendix A). 

Altogether our result show that intestinal FXR affects glucose-related genes in the duodenum as well as duodenal surface area in a stem-cell-proliferation-independent manner. However, glucose absorption kinetics are not regulated by intestinal FXR.

## 3. Discussion

Glucose absorption in the small intestine is a complex process and structural or functional alterations in the small intestine could lead to changes in glucose absorption rate and therefore in the magnitude of the postprandial glucose load [39]. In this study, we specifically focused on the potential role of intestinal FXR on glucose absorption by using iFXR-KO mice, and a pharmacological FXR agonist in combination with a novel approach to quantify intestinal glucose absorption in mice. Unexpectedly, based on our earlier studies in whole-body FXR^−/−^ mice, chow-fed iFXR-KO mice failed to show delayed glucose absorption, implying that effects in FXR^−/−^ mice likely have an extra-intestinal origin. This finding might be due to several reasons: first of all, whole-body FXR^−/−^ mice were generated by disrupting a 292-bp fragment of exon 2 in the *Fxr* gene [29,40], whereas the iFXR-KO mice have loxP sites flanking the last *Fxr* exon [41,42]. This difference might cause a variation in phenotype displayed by the two mouse models as was also previously recognised [40]. Secondly, the technical approach for analysis of glucose absorption was not equal. The novel glucose absorption test used in this study is based on two labelled glucose boli, a minute intravenous injection of a D-[6,6-^2^H_2_] glucose bolus and an oral administration of a D-[U-^13^C] glucose bolus, whereas our previous method applied continuous infusion of D-[6,6-^2^H_2_] glucose and an oral D-[U-^13^C] glucose bolus, a method that requires surgery prior to the test [29]. Apart from technical differences, *Fxr* ablation in liver was shown to have a major impact on intestinal gene expression as well as intestinal mucus barrier and this impact was not shown in iFXR-KO mice, indicating that some of the intestinal phenotypes of whole-body FXR^−/−^ mice have a hepatic origin [43]. These effects were however predominantly observed in the colon. Nevertheless, the delayed glucose absorption observed in whole-body FXR^−/−^ mice might be influenced by hepatic FXR ablation.

We also observed that pharmacological FXR activation did not significantly alter glucose absorption in wild-type mice, although a slight trend towards acceleration in glucose appearance rate was observed in the initial phase after bolus administration. Previous in vitro experiments have shown that the endogenous bile acid chenodeoxycholic acid (CDCA) induces glucose uptake in human intestinal epithelial cells, while blocking FXR inhibited this effect [44]. The authors speculated that increased GLUT2 transport from the nucleus to the cell surface is responsible for this FXR-mediated effect [44]. Importantly, non-steroidal FXR agonists may be able to activate FXR more strongly in the proximal part of the intestine [19], whereas endogenous steroidal ligands such as CDCA are absorbed more distally in the ileum. In our study, we measured several genes relevant for glucose absorption in the duodenum, which is the major region of glucose absorption in the small intestine [45]. Our iFXR-KO mice showed significantly higher *Glut2* expression in the duodenum. However, activation of FXR with GS3972 did not reduce this expression, suggesting that FXR does not have a direct impact on *Glut2* expression. As an additional regulatory mechanism, GLUT2 translocation is beyond transcriptional control: GLUT2 is predominantly located in the basolateral membrane, transporting glucose from the enterocyte into the blood stream but GLUT2 can rapidly translocate to the apical membrane upon glucose ingestion [46]. In addition, increased *Hk1* expression was observed in wild-type mice treated with an FXR agonist, as was paradoxically also previously shown for mice lacking FXR [29]. Hexokinase phosphorylates glucose to glucose-6-phosphate and glucose phosphorylation is an important process of a separate pathway for glucose absorption independent of GLUT2 [47]. Hexokinase also plays a major role in intestinal transcellular glucose transport, as it became evident from a study in GLUT2-deficient humans [48]. Importantly, the observed changes in gene expression of glucose transport-related genes due to either ablation or activation of FXR did not correlate with the kinetics of glucose absorption using a standard glucose load, indicating that there is absorptive overcapacity during standard feeding conditions as well as during the initial development of obesity as employed in our studies.

Another finding in this study is that FXR activation leads to increased villus length in the duodenum. Despite that these changes were relatively modest, they do translate into a more substantial increase in surface area, as villus length as well as width need to be multiplied to the power two to estimate surface area [49]. The increased villus length has been reported to be beneficial for the utilisation of nutrients, while shorter villi are associated with the presence of toxins or quicker renewal of villus cells [50]. Consistent with previous reports, HFD feeding resulted in increased villus length in wild-type mice and this might be due to adaptation to decreased motility and absorption capacity especially in the proximal intestine [51,52]. However, iFXR-KO mice did not show increased villus length in their duodenum to the same extend as wild-type mice. This indicates a slower or impaired adaptation in response to exposure to HFD in the absence of iFXR. The change in villus length in the duodenum was not accompanied by changes in total intestinal length, altered villus length in other intestinal sections or in total enterocyte mass. Despite the fact that the duodenum has substantially longer villi compared to the other segments (i.e., 500 µM in Figure 1f compared to 200–300 µM in Figure 1h), the murine jejunum itself is much longer than the duodenum and, together with the ileum, therefore accounts for the bulk of enterocyte mass [53]. Previous studies on the impact of intestinal FXR on villi morphology mainly focused on the ileum as it is the main site where endogenous FXR ligands are being absorbed [54]. One study has shown that iFXR-KO mice experienced severe morphological changes in the ileum such as shortening of villi and shrinkage of the submucosa and muscular layer upon ethanol administration [36]. Pharmacological FXR activation, on the other hand could improve the integrity of the ileum and protect against the deterioration of the epithelial barrier caused by biliary obstruction [18], cholestatic liver injury [55], or ischemia reperfusion injury [24]. These results illustrate the regulatory role of intestinal FXR in enteroprotection, which has been observed predominantly in disease models [56]. Indeed, previous studies have shown that activation of FXR primarily protects gut function by suppressing inflammation [25,57], which we did not observe in our model. Apart from the ileum, FXR was also detected in the duodenum, specifically in absorptive epithelial cells [58]. In our study, we observed that wild-type mice treated with an FXR agonist showed an increased villus length in the duodenum under one-week HFD, while iFXR-KO mice showed reduced villus length selectively in the proximal part of the small intestine compared to wild-type under various dietary conditions. This adaptation specifically in the duodenum in the absence of pre-existing damage suggests a different physiological role of FXR in the proximal intestine. Indeed, there are regional differences in identity along the gastrointestinal tract and nuclear receptors such as FXR have been proposed to play a regulatory role, thus far predominantly in the more distal areas [59].

The intestinal epithelium contains different types of cells including enterocytes, goblet cells and endocrine cells and villus or crypt length can be influenced by proliferation rate and turnover rate of intestinal epithelial cells [60]. Activation of FXR has been shown to inhibit intestinal epithelial cell proliferation in in vitro experiments [61,62] and deficiency of FXR promoted cell proliferation and inflammation in the colon [63]. In our study, long-term HFD also led to increased ileal crypt length, suggesting that after prolonged obesogenic conditions and metabolic derangements additional FXR-regulated effects on intestinal morphology may become apparent. During the early stage of obesity development, we did not observe a significant difference in proliferation rate of new enterocytes in the duodenum between wild-type and iFXR-KO mice, treated or untreated with an FXR agonist. Thus, the increased villus length found in duodenum of wild-type mice treated with the FXR agonist must be due to a longer life span of epithelial cells. Indeed, a previous study has shown that FXR activation increases cell survival and protects against apoptotic cell death of human and mouse gastric epithelial cells by targeting antiapoptotic proteins such as keratin 13 [64]. Interestingly, HK1 was found to be involved in the transduction of cell death signals and it counters tumour necrosis factor (TNF)-induced apoptosis [65,66]. Therefore, the increased expression of *Hk1* in the duodenum of mice treated with an FXR agonist might also contribute to the life span of epithelial cells, which may subsequently lead to increased villus length in those mice.

Unexpectedly, we did not observe beneficial effects on NAFLD development, insulin resistance or glucose homeostasis in iFXR-KO mice after 10-week HFD feeding, as was observed in previous studies [21,22,32]. Western type diets and subsequent weight gain are the major driver of these features of the metabolic syndrome and the iFXR-KO mice did indeed gain relatively less weight compared to controls in these studies. Importantly, we did not observe blunted weight gain in iFXR-KO mice under obesogenic conditions. In line with our findings, a recent study on the role of FXR in the beneficial effects of vertical sleeve gastrectomy also observed weight gain at a similar pace in iFXR-KO mice compared to wild-type mice on HFD prior to the surgical intervention [12]. Unlike earlier studies, these mice received a diet identical to the one applied in our studies, suggesting that dietary factors beyond fat content alone may modulate weight gain in iFXR-KO mice. The beneficial effects on weight gain in earlier studies were indeed also dependent on the microbiota, which is a strongly modified by diet [67,68]. Interestingly, rather than an effect on intestinal energy metabolism, the beneficial effects of the vertical sleeve gastrectomy were attributed to an FXR-dependent effect on bile acid composition and subsequent reduced fat uptake that was blunted in global FXR^−/−^ mice, yet retained in liver- or intestine-specific FXR-KO models [12]. 

We also acknowledge some limitations of this study. We did not study the effect of long-term pharmacological FXR stimulation on glucose absorption or its application under pathophysiological conditions, i.e., during later stages of obesity. This could provide more insight on how FXR agonism affects insulin resistance-induced processes that appear later in the disease process. We also limited the experiments to male mice, as was also the case in our study on global FXR^−/−^ mice [29] 

In conclusion, we studied the role of intestinal FXR on glucose absorption and morphology of the small intestine under chow, short- and long-term HFD feeding conditions, by using iFXR-KO mice and the FXR agonist GS3972. Our results show that iFXR-KO mice do not present with delayed glucose absorption as whole-body FXR^−/−^ mice do, despite decreased hexokinase expression and increased duodenal surface area. The changes in intestinal structure are limited to the duodenum, which might be due increased epithelial cell turnover in iFXR-KO mice, a process that may be delayed by FXR activation. 

## 4. Materials and Methods

### 4.1. Mice

The intestine-specific FXR^−/−^ mice were generated by cross-breeding of mice containing Cre-recombinase under the control of the villin promoter [69] with Fxr-floxed mice [42], kindly provided by Dr Frank J Gonzalez (NIH, Bethesda, MD, USA). Littermates harbouring the loxP sites (flox/flox mice) were used as controls. Ten- to sixteen-week-old male iFXR-KO and control mice were housed individually in a light (12:12) and temperature (21 °C)-controlled facility and received chow or high-fat diet (HFD) (60 kcal %, D17041409) and water ad libitum. Animals were fasted for 4 h and then terminated under anaesthesia. Another cohort of male iFXR-KO and wild-type mice received one-week HFD combined with the FXR agonist GS3972 (kindly provided by Gilead Inc., Foster City, CA, USA) dissolved in 0.5% methylcellulose (10 mg/kg) or methycellulose control solution that was orally administered daily for 6 days. After the last gavage, mice were fasted for 4 h and terminated under anaesthesia. During the termination, organs and tissues were carefully dissected, weighed and stored at −80 °C before use. All experimental procedures were approved by the review board of the Animal Care and Use Committee of the Groningen University in accordance with local regulations for use of experimental animals. 

### 4.2. Glucose Absorption Test

Mice were fasted for 9 h and D-[6,6-^2^H_2_] glucose (150 mg/kg) was injected intravenously (IV), while a glucose bolus containing 1 g/kg glucose (970 mg/kg glucose and 30 mg/kg D-[U-^13^C] glucose was administered orally directly thereafter. Blood glucose levels were measured from the tail-tip at several time points after glucose administration (0, 15, 30, 45, 60, 90 and 120 min) using a handled glucometer (Accu Check Advantage, Roche Diabetes Care, Ind., USA). Bloodspots were collected on filter paper to assess tracer enrichment in the blood circulation at these time points. At the time points of 0, 5, 15 and 120 min, additional bloodspots were collected for insulin measurement using the ultrasensitive rat ELISA kits and insulin mouse standard (#90060 and #90020, Crystal Chem Inc., Zaandam, The Netherlands). The insulin assay was performed according to the manufacturer’s instruction. 80 °C before use. 

### 4.3. Gas Chromatography-Mass Spectrometry

Glucose was first extracted by ethanol from blood spots and converted to the glucose pentacetate derivative using pyridine and acetic anhydride. The solution was evaporated at 60 °C under a stream of nitrogen and the residue was dissolved in ethyl acetate, which was transferred into injection vials for analysis by gas chromatography-mass spectrometry (GC/MS). The derivatives were separated on a Zebron-1701 30 m × 0.25 mm ID (0.25 µm film thickness) capillary column. Mass spectrometry analyses were performed by positive chemical ionisation with ammonia. Ions monitored; *m*/*z* 408–414 (m0–m6). 

### 4.4. Tracer Kinetics

Fractional distributions of glucose isotopologues as measured by GC/MS (m0–m6) were first adjusted for natural abundance of ^13^C atoms (M0–M6). The calculation of different parameters (tracer concentrations at different time point, glucose clearance rate, glucose disposal rate, glucose appearance rate, glucose recovery at different time points, etc.) was based on formulas according to van Dijk et al. [70] (Appendix A). The “Wagner–Nelson method”, a pharmacokinetic approach, was used to estimate absorption over time [71]. Further details on the novel approach are provided in Appendix A.

### 4.5. Histology

Intestinal and liver samples were fixed in 4% neutral buffered formaldehyde, embedded in paraffin, cut into 4-μm sections, and stained with haematoxylin/eosin (H&E) by standard procedures. Images were obtained using a Hamamatsu NanoZoomer (Hamamatsu Photonics, Almere, The Netherlands) and viewed on ImageScope Viewer software (V11.2.0.780 Aperio). 10 random complete villi and crypts in one section of small intestine (duodenum, jejunum and ileum) from each mouse were chosen to measure the villus length and crypt length, and the average was calculated. Crypt length was defined as the invagination between two villi and villus length was defined as the distance between the tips of the villi to the crypt transition [72]. 

### 4.6. Fluorescence Staining

Brdu (5-bromo-2’-deoxyuridine, 100 mg/kg in PBS) was injected intraperitoneally to iFXR-KO mice and wild-type mice 24 h before termination. Duodenal samples were collected, fixed in 4% neutral buffered formaldehyde, embedded in paraffin and cut into 4-μm sections. Paraffin sections were prepared for classic immunohistochemistry (IHC) staining by deparaffinisation in xylene and rehydrating through an ethanol series to distilled water. Tissues were permeabilised for 10 min in 0.2% Triton X-100 in PBS. Antigen retrieval was performed in citrate buffer (pH 6.0) in water bath at 95 °C for 30 min. Slides were then blocked in a humidity chamber in 1% bovine serum albumin (BSA) in PBS for 1 h, incubated with primary antibody Rat anti-BrdU (1:300) (Accurate Chemical #OBT0030G) at 4 °C overnight, washed in 0.2% Triton X-100 in PBS, incubated with secondary antibody Goat anti-Rat Alexa Fluor^®^ 488 (1:200) for 1 h, washed in 0.2% Triton X-100 in PBS, stained with DAPI (4′,6-diamidino-2-phenylindole), and mounted with Prolong Gold (Invitrogen). Images from the whole sections of duodenum from each mouse were obtained using high content fluorescence Widefield microscopy TissueFaxs (TissueGnostics GmbH, Vienna, Austria). Single field images of the duodenum sections were obtained with a Leica DM LB fluorescence microscope (Leica, Wetzlar, Germany). 

### 4.7. Plasma Citrulline Measurement 

Blood was collected from the tail vein into EDTA-treated haematocrit capillary tubes. After centrifugation at 8000 g for 10 min at 4 °C, plasma was collected for citrulline determination with the automated ion-exchange column chromatography method as previously described [73,74].

### 4.8. Real-Time Quantitative Reverse Transcription-PCR 

Real-time quantitative reverse transcription-PCR was performed to analyse the mRNA expression of genes relevant for glucose absorption in iFXR-KO and control mice. Five segments of intestine (from proximal intestine to distal intestine: 0, 20, 40, 80, 100%) and liver were collected for RNA isolation. Total RNA was isolated from intestine or liver samples using TRI reagent (Sigma-Aldrich, St. Louis, MO, USA). cDNA was synthesised by reverse transcription using moloney murine leukaemia virus reverse transcriptase and random nonamers according to the manufacturer’s protocol. mRNA expression levels were analysed using the StepOne Real-Time PCR system (Applied Biosystems Europe, Nieuwerkerk aan Den IJssel, The Netherlands). PCR results were normalised to 36B4 mRNA levels. Primers and probe sequences are listed in Appendix A.

### 4.9. Bile Acid Profile in Plasma and Gallbladder

Bile acids in plasma and gallbladder were measured by ultra (U)HPLC-MS/MS on a Nexera X2 UHPLC system (Shimadzu, Kyoto, Japan), coupled to a SCIEX QTRAP 4500 MD triple quadrupole mass spectrometer (SCIEX, Framingham, MA, USA) [75]. Bile acid concentration in plasma was measured in 25 μL of homogenised plasma. An internal standard containing D4-cholate, D4-chenodeoxycholate, D4-glycocholate, D4-taurocholate, D4-glycochenodeoxycholate, and D4-taurochenodeoxycholate was used for quantification. Concentration of different bile acids from plasma and gallbladder in WT and iFXR-KO mice on 10-week HFD was shown in Appendix A. 

### 4.10. Statistical Analysis

Significance of differences between two groups were tested by non-parametric Mann–Whitney *U* test, whereas significance of difference among multiple groups was assessed by Kruskal–Wallis H-test, followed by post-hoc Conover analysis test using the GraphPad Prism 8.00 software package (GraphPad Software, San Diego, CA, USA). Significance was indicated as * *p* < 0.05, ** *p* < 0.01, *** *p* < 0.001.

## Figures and Tables

**Figure 1 ijms-24-04132-f001:**
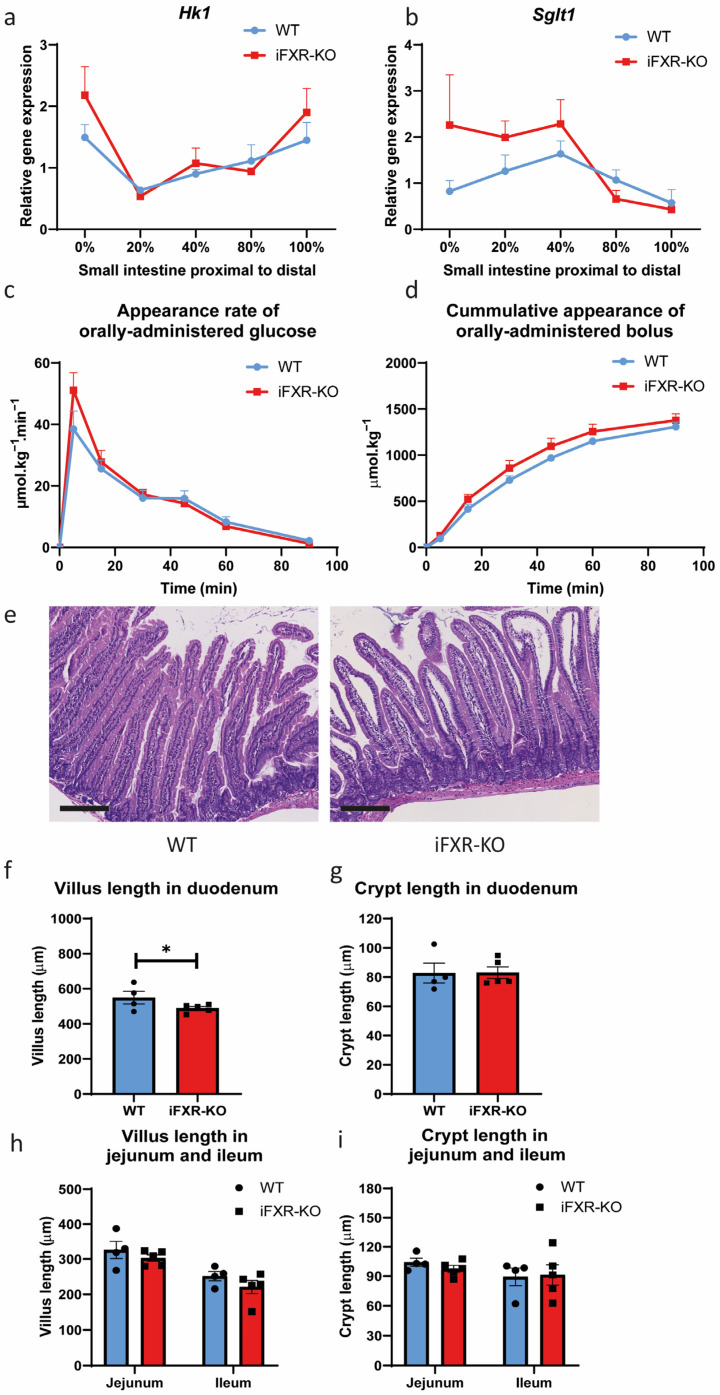
FXR deficiency in intestine does not lead to delayed glucose absorption appearance. (**a**,**b**) Quantitative real-time PCR of glucose absorption gene expression (*Hk1* and *Sglt1*) in 5 parts of intestine (0, 20, 40, 80 and 100% of small intestine from proximal to distal) from wild-type (WT) and intestine-specific FXR^−/−^ (iFXR-KO) mice on chow diet; (**c**) Orally administrated D-[U-^13^C] glucose appearance rate, (**d**) recovery of orally administrated glucose bolus at 0, 5, 15, 30, 45, 60 and 90 min during glucose absorption test from WT and iFXR-KO mice on chow diet; (**e**) Representative image of H&E staining of duodenum from WT and iFXR-KO mice on chow diet (magnification 10×; bar represents 200 µm); (**f**) Quantified analysis of villus length and (**g**) crypt length of duodenum from WT and iFXR-KO mice on chow (10 villi were selected from each mouse for qualification); (**h**) Quantified analysis of villus length and (**i**) crypt length of jejunum and ileum from WT and iFXR-KO mice on chow diet (10 villi were selected from each mouse for qualification). Statistical significance was determined using the Mann–Whitney test in WT and iFXR-KO mice. All panels: n = 4–5/group. Data are represented as mean ± SEM, Panel (**f**): * *p* < 0.05.

**Figure 2 ijms-24-04132-f002:**
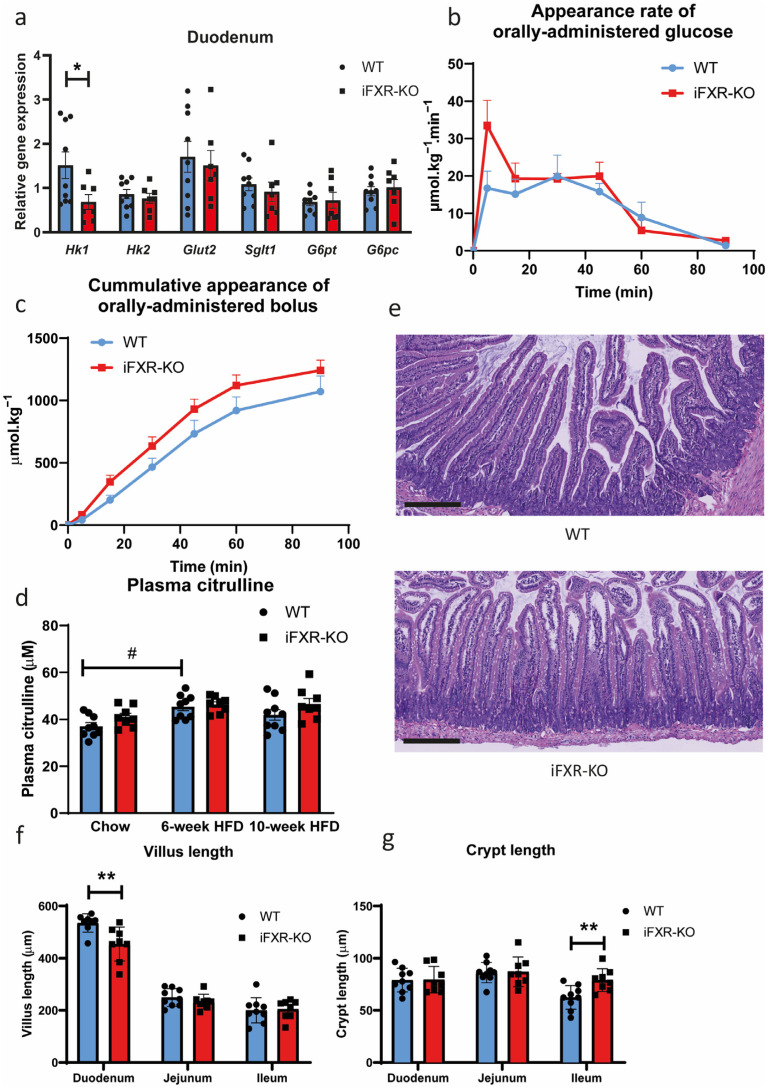
Long-term HFD exposure does not lead to changes of glucose absorption in iFXR-KO mice (**a**) Quantitative real-time PCR of glucose absorption gene expression in duodenum from wild-type (WT) and intestine-specific FXR^−/−^ (iFXR-KO) mice after 10-week high-fat-diet (HFD); (**b**) Orally administrated D-[U-^13^C] glucose appearance rate and (**c**) recovery of orally administrated glucose bolus at 0, 5, 15, 30, 45, 60 and 90 min during glucose absorption test from WT and iFXR-KO mice after 10-week HFD; (**d**) Plasma citrulline level from WT and iFXR-KO mice when they were on chow diet, 6-week HFD and 10-week HFD; (**e**) Representative image of H&E staining of duodenum from WT and iFXR-KO mice after 10-week HFD (magnification 10×; bar represents 200 µm); (**f**) Quantified analysis of villus length and (**g**) crypt length of duodenum, jejunum and ileum from WT and iFXR-KO mice after 10-week HFD (10 villi were selected from each mouse for qualification). Statistical significance was determined using the Mann–Whitney test in WT and iFXR-KO mice. All panels: n = 8–9/group, data are represented as mean ± SEM, panels (**a**,**f**,**g**): * *p* < 0.05 and ** *p* < 0.01 when iFXR-KO mice compared with WT mice; panel (**d**): # *p* < 0.05 when WT mice on chow condition compared with HFD condition.

**Figure 3 ijms-24-04132-f003:**
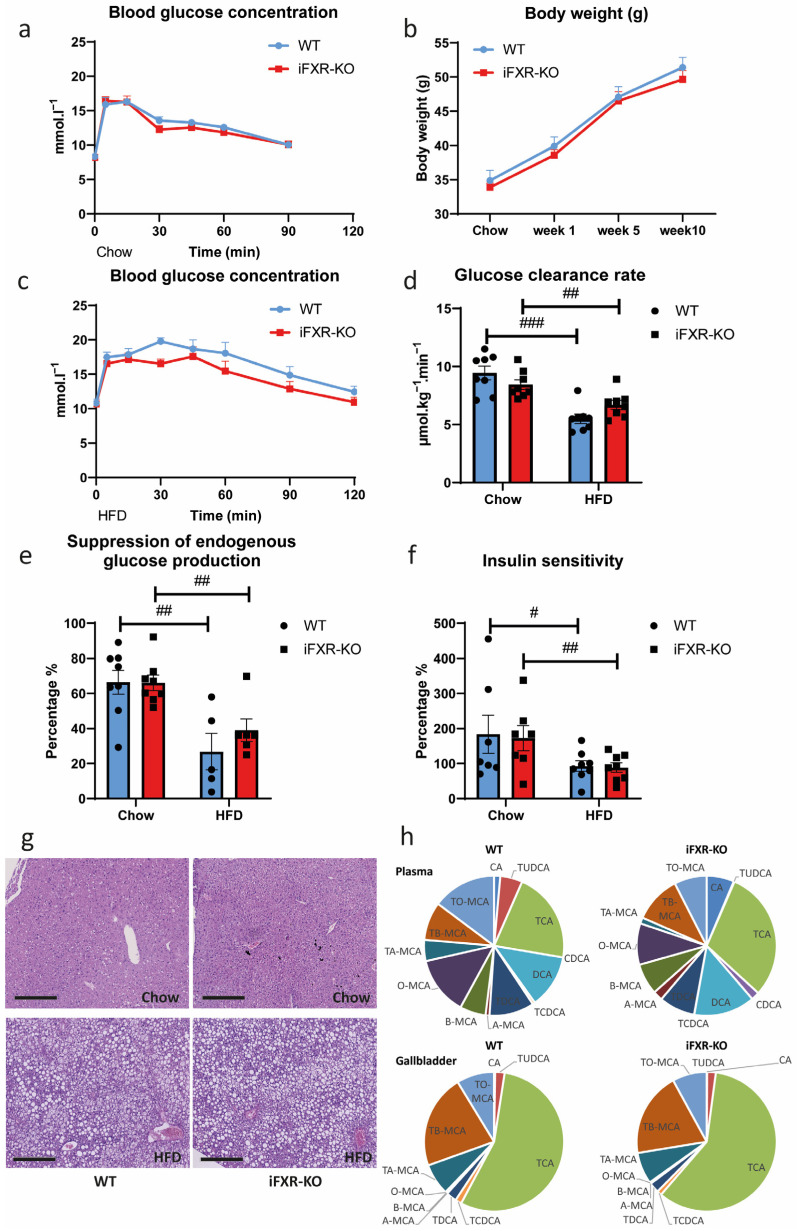
Long-term HFD exposure leads to impaired glucose metabolism in both WT and iFXR-KO mice. (**a**) Blood glucose concentration at 0, 5, 15, 30, 45, 60 and 90 min during glucose absorption test of wild-type (WT) and intestine-specific FXR^−/−^ (iFXR-KO) mice on chow diet; (**b**) Body weight change of WT and iFXR-KO mice during the period of 10-week high-fat-diet (HFD) feeding; (**c**) Blood glucose concentration at 0, 5, 15, 30, 45, 60, 90 and 120 min during glucose absorption test of WT and iFXR-KO mice after 10-week HFD; (**d**) Glucose clearance rate, (**e**) hepatic glucose suppression and (**f**) insulin sensitivity of WT and iFXR-KO mice during glucose absorption test on chow and on 10-week HFD; (**g**) Representative image of H&E staining of liver from WT and iFXR-KO mice on chow or on 10-week HFD (magnification 8x; bar represents 300 µm); (**h**) Relative bile acid composition in plasma and gallbladders from WT and iFXR-KO mice on 10-week HFD. CA: cholic acid, (T)CDCA: (tauro)chenodeoxycholic acid, TUDCA: tauroursodeoxycholic acid, (A,B,O)MCA: (α,β,ω)muricholic acid, TMCA: tauromuricholic acid, TCA: taurocholic acid, DCA: deoxycholic acid. Statistical significance was determined using the Mann–Whitney test in WT and iFXR-KO mice. All panels: n = 6–9/group, data are represented as mean ± SEM, # *p* < 0.05, ## *p* < 0.01 and ### *p* < 0.001 when WT mice on chow condition compared with HFD condition.

**Figure 4 ijms-24-04132-f004:**
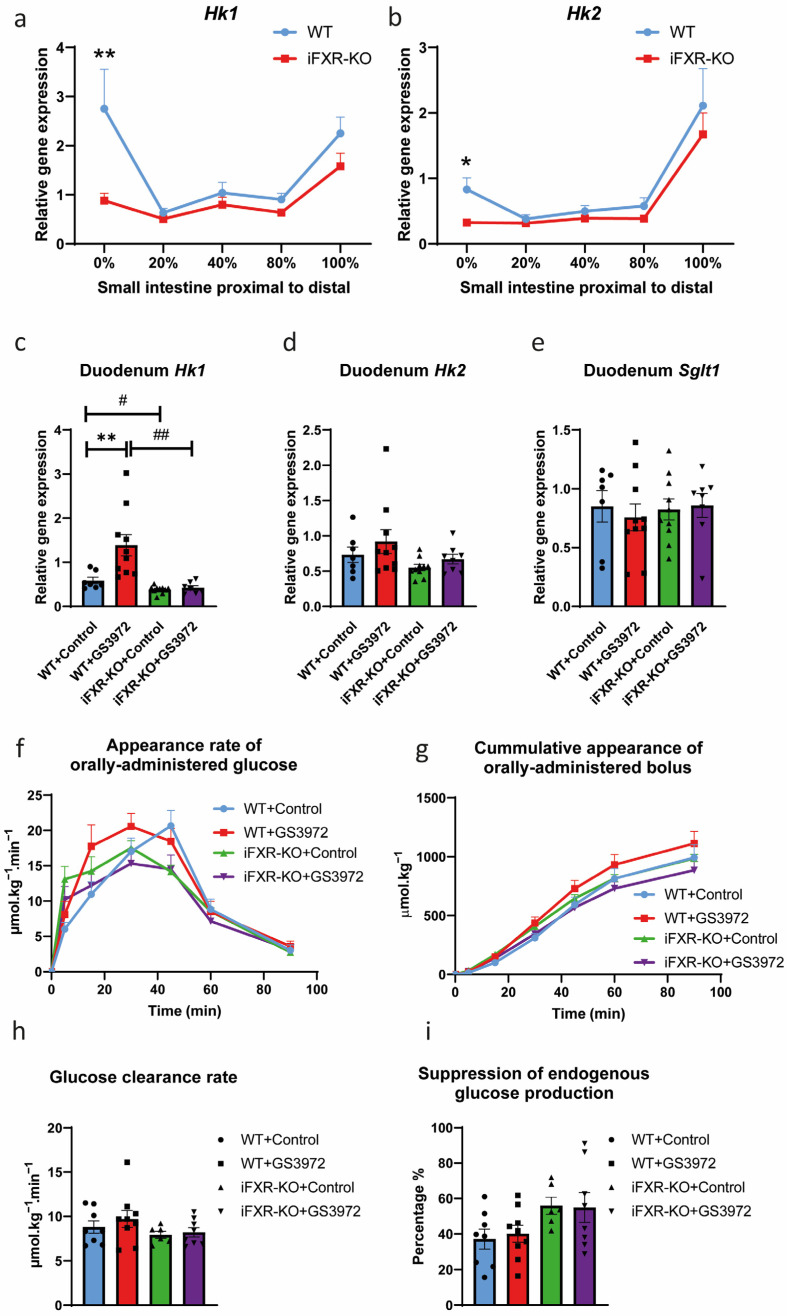
Activation of FXR increased *Hk1* expression in duodenum. (**a**,**b**) Quantitative real-time PCR of glucose absorption gene expression (*Hk1* and *Hk2*) in 5 parts of intestine (0, 20, 40, 80 and 100% of small intestine from proximal to distal) from wild-type (WT) and intestine-specific FXR^−/−^ (iFXR-KO) mice on one-week high-fat-diet (HFD) chow diet; (**c**–**e**) Quantitative real-time PCR of glucose absorption gene expression (*Hk1*, *Hk2* and *Sglt1*) in duodenum from WT and iFXR-KO mice treated with or without GS3972 for one week under HFD condition; (**f**) Orally administrated D-[U-^13^C] glucose appearance rate and (**g**) recovery of orally administrated glucose bolus at 0, 5, 15, 30, 45, 60 and 90 min during glucose absorption test from WT and iFXR-KO mice treated with or without GS3972 for one week under HFD; (**h**) Glucose clearance rate and (**i**) hepatic glucose suppression from WT and iFXR-KO mice treated with or without GS3972 for one week under HFD. Panels (**a**,**b**): Statistical significance was determined using the Mann–Whitney test in WT and iFXR-KO mice. n = 5/group, data are represented as mean ± SEM, * *p* < 0.05 and ** *p* < 0.01. Panels (**c**–**i**): Statistical significance was determined using Kruskal–Wallis H-test followed by post-hoc Conover pairwise comparisons. n = 7–9/group. Data are represented as mean ± SEM, * *p* < 0.05 and ** *p* < 0.01 when mice with GS3972 treatment compared with control, # *p* < 0.05 and ## *p* < 0.01 when iFXR-KO mice compared with WT mice.

**Figure 5 ijms-24-04132-f005:**
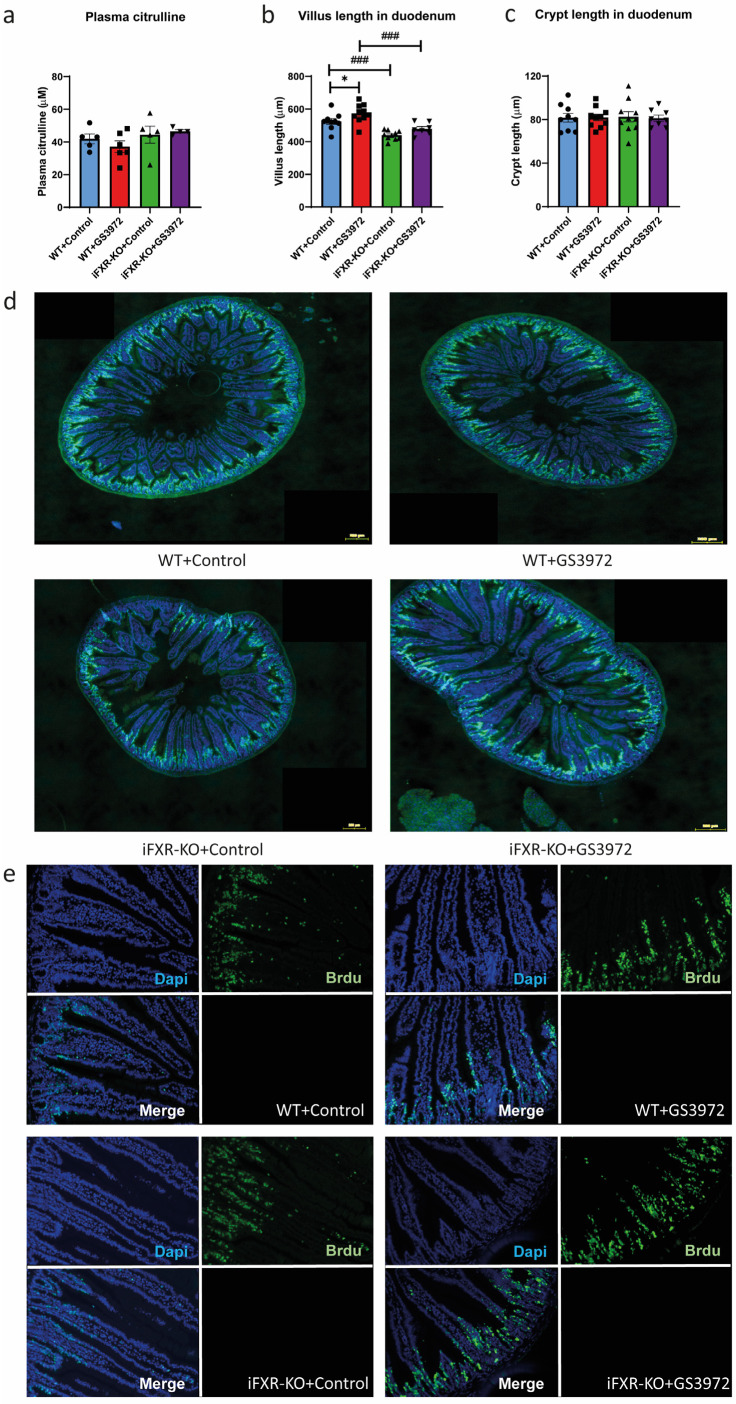
Activation of FXR does not lead to increased proliferation rate of epithelial cells in duodenum (**a**) Plasma citrulline level from WT and iFXR-KO mice wild-type (WT) and intestine-specific FXR^−/−^ (iFXR-KO) mice treated with or without GS3972 for one week under high-fat-diet (HFD) condition. (**b**) Quantified analysis of villus length and (**c**) crypt length of duodenum from WT and iFXR-KO mice treated with or without GS3972 for one week under HFD (10 villi were selected from each mouse for qualification). (**d**,**e**) Representative image of immunofluorescence staining for the proliferative cell marker Brdu in duodenum from WT and iFXR-KO mice treated with or without GS3972 for one week under HFD. (**d**) Image of whole section of duodenum captured by TissueFax (magnification 10×; bar represents 200 µm). (**e**) Image of part section of duodenum captured by the digital camera (magnification 16×). Statistical significance was determined using Kruskal–Wallis H-test followed by post-hoc Conover pairwise comparisons. Panel (**a**): n = 4–6/group; Panel (**b**–**e**): n = 7–9/group. Data are represented as mean ± SEM, * *p* < 0.05 when mice with GS3972 treatment compared with control, ### *p* < 0.001 when iFXR-KO mice compared with WT mice.

## Data Availability

Not applicable.

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
