# Peer review of "Intestinal Farnesoid X Receptor Modulates Duodenal Surface Area but Does Not Control Glucose Absorption in Mice"

_ijms, 2023, doi:10.3390/ijms24044132_

Round 1

Reviewer 1 Report

I inform you that I reviewed the submitted manuscript (Manuscript ID: ijms-2147259).  The re-review are required.

The author indicated that the delayed glucose absorption reported in whole-body FXR-/- mice was not seen in intestine-specific FXR-/- mice. I believe that the study are interesting in investigating molecular interaction of the farnesoid X receptor (FXR) system in intestinal absorption of lipid, cholesterol, and glucose. But, there are serious problems that should be addressed before this paper is accepted for publication for your journal.

Conditional gene knockout represents an extremely powerful approach to study the function of single genes in the liver system as well as intestine system. Using the loxP/loxP-AlbCre-induced conditional knockout system, you should generated a mouse that lacks FXR specifically in the liver. There may be a link between the intestine system and the liver system.

Author Response

We thank the reviewer for a thorough assessment of our manuscript and the generally positive comments on the major findings of our manuscript entitled “Intestinal Farnesoid X Receptor modulates duodenal surface area but does not control glucose absorption in mice“. The attached document addresses the issue raised by the reviewers and edited the manuscript accordingly. We believe it is now suitable for publication in International Journal of Molecular Sciences.

Reviewer 2 Report

  1. I recommend moving Figs S1 and S3 to the main manuscript. Following the results in these figures (and Fig 1 &4) is hard; the readers have to jump back and forth between the various panels of the regular figures and supplemental figures.
  2. Why did the authors choose GS3972? Why did the authors choose a 10 mg/kg dose of GS3972? Please justify the choice over other FXR agonists.
  3. Why did the authors choose male mice for these experiments? Do the authors expect similar results in female mice? Please justify the choice of using only male mice in these studies.
  4. Please add a reference to line 289 (…with earlier results…)
  5. The method of individual BA quantification is missing from the Materials and methods section. Please describe the applied method briefly. Also, in a supplementary table, please list the absolute concentrations of individual bile acids quantified in plasma and gallbladder bile. 

Author Response

We thank the reviewer for a thorough assessment of our manuscript and the generally positive comments on the major findings of our manuscript entitled “Intestinal Farnesoid X Receptor modulates duodenal surface area but does not control glucose absorption in mice“. Below we address the issues raised by the reviewers and edited the manuscript accordingly. We believe it is now suitable for publication in International Journal of Molecular Sciences.

Reviewer 3 Report

The manuscript “Intestinal Farnesoid X Receptor modulates duodenal surface area but does not control glucose absorption in mice” by Jiufang Yang, et al. applied a dual-label glucose kinetic approach in intestine-specif FXR-/- mice to study the role of FXR in glucose absorption. They found that the delayed glucose absorption in whole-body FXR-/- mice is not due to absence of intestinal FXR, but intestinal FXR does have a role in small intestinal surface area. The topic is interesting, and the manuscript is well written. The experimental design is reasonable, and the results are clear presented. I recommend publish this manuscript in IJMS after fix the following minor comments:

1.     A lot of the raw data points are very different from mean value, for example, the raw data points in Fig 2a or Fig 4e. I am not familiar with common error bar in this measurement. Can the authors comment on why some of the data points are so different from mean? And potential way to improve the sampling?

2.     In Fig 1f, the authors claim there is significant difference between the blue bar and red bar, I agree with this statement. However, the mean of blue and red bar only differ by about several tens of um (I can’t find the value of difference in the main text). Can the authors expand the discussion on the consequence of this small difference? Even though it’s correct to say that they make statistic difference, but how different are the consequence in mice?

3.     Similar to point 2, in page 11 line 285, the authors claim the morphology of the duodenum was slightly but significantly affected by GS3972-induced FXR activation. As a general audience who is not familiar with the background, I want to ask why should I care about this slightly effect? Some further discussions on the consequences of these slightly but significantly effects are appreciated.

Author Response

(The authors gave the same response as above.)
